# Environmental Monitoring of Tritium ($^3$H) and Radiocarbon ($^{14}$C) Levels in Mafikeng Groundwater Using Alpha/Beta Spectrometry

Joseph Mathuthu *, Omphile Edna Dzimba and Manny Mathuthu

Center for Applied Radiation Science and Technology (CARST), North-West University (Mafikeng Campus), Mmabatho 2735, South Africa; omphileedna@gmail.com (O.E.D.); manny.mathuthu@nwu.ac.za (M.M.)
* Correspondence: mp18797@gmail.com; Tel.: +27-76-920-5584

**Abstract:** With the current state of water scarcity in Mafikeng, South Africa, due to low water levels and an increasing population, it is therefore crucial to ensure the sustainability and availability of the existing water resources. In this study, the levels of tritium and radiocarbon in groundwater (boreholes) at selected villages in Mafikeng were determined using the Perkin Elmer Ultra Low Liquid Scintillation Counter 2000. The mean activity concentrations of tritium are 3.61304 ± 0.00612 Bq/L and 3.86014 ± 0.00739 Bq/L for samples from Dibate and Lokaleng villages, respectively, whereas 1.83392 ± 0.02265 Bq/L is for Moletsamongwe, Lekung, Airport View and Seweding. Moreover, the mean activity concentrations of radiocarbon from Dibate and Lokaleng are 0.59296 ± 0.00886 Bq/L and 0.8333 ± 0.0126 Bq/L, respectively, whereas for, Moletsamongwe, Lekung, Airport View and Seweding, they are 1.3752713 ± 0.01968 Bq/L. Two (2) out of the forty (40) samples analysed for radiocarbon are below the minimum detectable activity of 0.33627 Bq/L. The average annual effective dose (AED) of tritium for analysed samples from Dibate and Lokaleng villages are 0.04754 μSv/y and 0.05079 μSv/y, respectively, whereas it is 0.02413 μSv/y for Moletsamongwe, Lekung, Airport View and Seweding. The average AED for radiocarbon is 0.251404 μSv/y and 0.36604 μSv/y for samples from Dibate and Lokaleng, respectively, whereas it is 0.58309 μSv/y for Moletsamongwe, Lekung, Airport View and Seweding village. The evaluated lifetime cancer risk for mortality and morbidity in adults is lower than the radiological cancer risk limit of $1.63 \times 10^{-3}$ set by regulatory agencies; hence, the consumption of the studied groundwater from the selected villages will not pose any health risks in terms of tritium and radiocarbon levels.

**Keywords:** groundwater; radioactivity; lifetime cancer risk; cancer mortality and morbidity; annual effective dose; human consumption; tritium; radiocarbon; Mafikeng

## 1. Introduction

With the rapid increase in South Africa's population, the need for clean water is escalating. Some places in the North West province of South Africa are already facing the issue of clean water shortages because of extreme weather conditions such as warmer temperatures and a lack of rainfall [1]. Hence, the North West Province obtains most of its water from groundwater resources. There are global concerns that the usage of water above permitted limits could result in dire health consequences, and in order to address this concern, water samples should be analysed against a range of health- and non-health-based physico-chemical standards. Most of these standards are based on those of the World Health Organization (WHO) [2].

The availability of water is essential for the survival and development of all living organisms, including humans. Its indispensability extends to all aspects of human life, making it a natural gift that cannot be overlooked in any habitat. Water is important for life on Earth; therefore, having safe and clean water should be the main concern for sustainable

development. Groundwater is a great resource for drinking water, but it appears to be vulnerable to radionuclide pollution. Most radionuclides are of natural origin; however, there is also groundwater pollution from artificial medical radio tracers improperly disposed of in the environment. Groundwater is also often withdrawn for agricultural, municipal and industrial uses by constructing and operating extraction wells; however, these human activities are also the main factors contributing to groundwater pollution [3]. Industrial discharges containing heavy metals, solvents, petroleum products and/or radioactive materials pose a notable risk to the quality of underground water. Improper storage and/or disposal of hazardous radioactive waste substances can lead to soil contamination and gradually seep into underlying aquifers. The use of fertilizers, pesticides and herbicides in agricultural practices can infiltrate the soil and reach groundwater through a process called leaching. When applied inappropriately, they become potential groundwater pollutants. Improper municipal waste disposal can allow the leachate to seep into groundwater, containing various contaminants, including radionuclides. Furthermore, mining activities, such as Uranium mining and milling, which are common here in the North-West Province of South Africa, can also contribute to groundwater pollution. Groundwater can also be polluted through the discharge of contaminated water, leaching from tailings and/or seepage from storage ponds.

There are radionuclides used as environmental tracers that are produced by cosmic-ray spallation in the atmosphere. However, some of these nuclides, such as carbon-14 ($^{14}$C) and tritium ($^{3}$H), are also produced by human activities, such as mining, nuclear weapons testing, industrial discharges, etc., [4]. Both qualitative and quantitative data on recharge mechanisms allow for the monitoring of changes as future activities occur [5].

Environmental tracers make it possible to assess the internal dynamics of a groundwater system and to study the quantity of timescales associated with groundwater flow [6,7]. The importance of understanding and characterizing the sustainability of groundwater as a water supply source for the human race is increasing along with population growth and climate change. The interpretation of environmental tracers can provide an integrated estimate of the flow velocity between two given locations, for instance, between a recharge zone and a discharge zone or between a contaminant source and potential receptors [8]. The most direct environmental tracers for groundwater dating are tritium ($^{3}$H) and radiocarbon ($^{14}$C) [9]. The radiocarbon and tritium content in groundwater can be used to study the residence time of the groundwater in the aquifer [10].

Naturally, on Earth, the majority of the radiocarbon is produced in the upper atmosphere. The upper atmosphere is continually being attacked by high-energy cosmic rays that are coming from outside the solar system. These high-energy particles proceed through a series of nuclear processes, producing some neutrons with a sluggish motion [11]. The impact of burning fossil fuels is the other significant artificial factor important to radiocarbon dating. This causes a significant volume of carbon dioxide to be released into the atmosphere, which essentially contains no radiocarbon (because the organic material from which it derives is so ancient). This lowers the atmospheric radiocarbon-to-stable carbon ratio. The fossil fuel or "Suess" effect is the name given to this impact. It is very challenging to radiocarbon date objects from the years 1650 to 1950, in part because of this [11].

The aim of this research was to ascertain whether the groundwater in Mafikeng is radiologically safe for drinking. Hence, the objectives were, firstly, to measure the activity concentrations of $^{3}$H and $^{14}$C using alpha/beta spectrometry and estimate the annual effective dose rates (AED) due to the isotopes in the groundwater collected from the various villages in Mahikeng. The AED gives the annual whole-body dose rate from ingestion of the analysed water samples. It is estimated as a product of three factors: the water consumption rate (L/y), the $^{3}$H and $^{14}$C activity concentrations in the water (Bq/L), and the dose rate conversion factor (e.g., μSv/Bq) for the $^{3}$H and $^{14}$C. Secondly, to estimate the lifetime cancer risk (LTCR) mortality and morbidity for adult males and females due to these isotopes. The result of this study will reveal the extent of pollution of groundwater (boreholes) by $^{14}$C and $^{3}$H in the rural villages in Mafikeng.

## 2. Materials and Methods

### 2.1. Geological and Hydrological Settings

The research was conducted using samples from four selected villages in North West Province, South Africa (Figure 1). Generally, the geology of the area comprises nine lithological stratigraphic classes—namely, Swazian, Kalahari, Dwyka, Malmari, Allanridge, Kraaipan, Kameeldooms, Blackreef and Klipriviersberg. The Swazian category, which consists of granite and gneiss, is the dominant geological structure in the study area. These are very important in increasing the occurrence of groundwater due to the fractures present. The Malmari is made up of limestone, subordinate chert, minor carbonaceous shale, quartzite and dolomite. The chert-rich formations are significant in the formation of groundwater's main aquifers. In the Blackreef, some fractures experience low permeability. The Kalahari is the most dominant geological landscape, with superficial deposits consisting of gravels, clays, sandstone, silcrete, calcrete and aeolian sand. Groundwater occurs mostly in crystalline rocks that are weathered near the surface and are later siphoned to fractured zones in deeper portions.

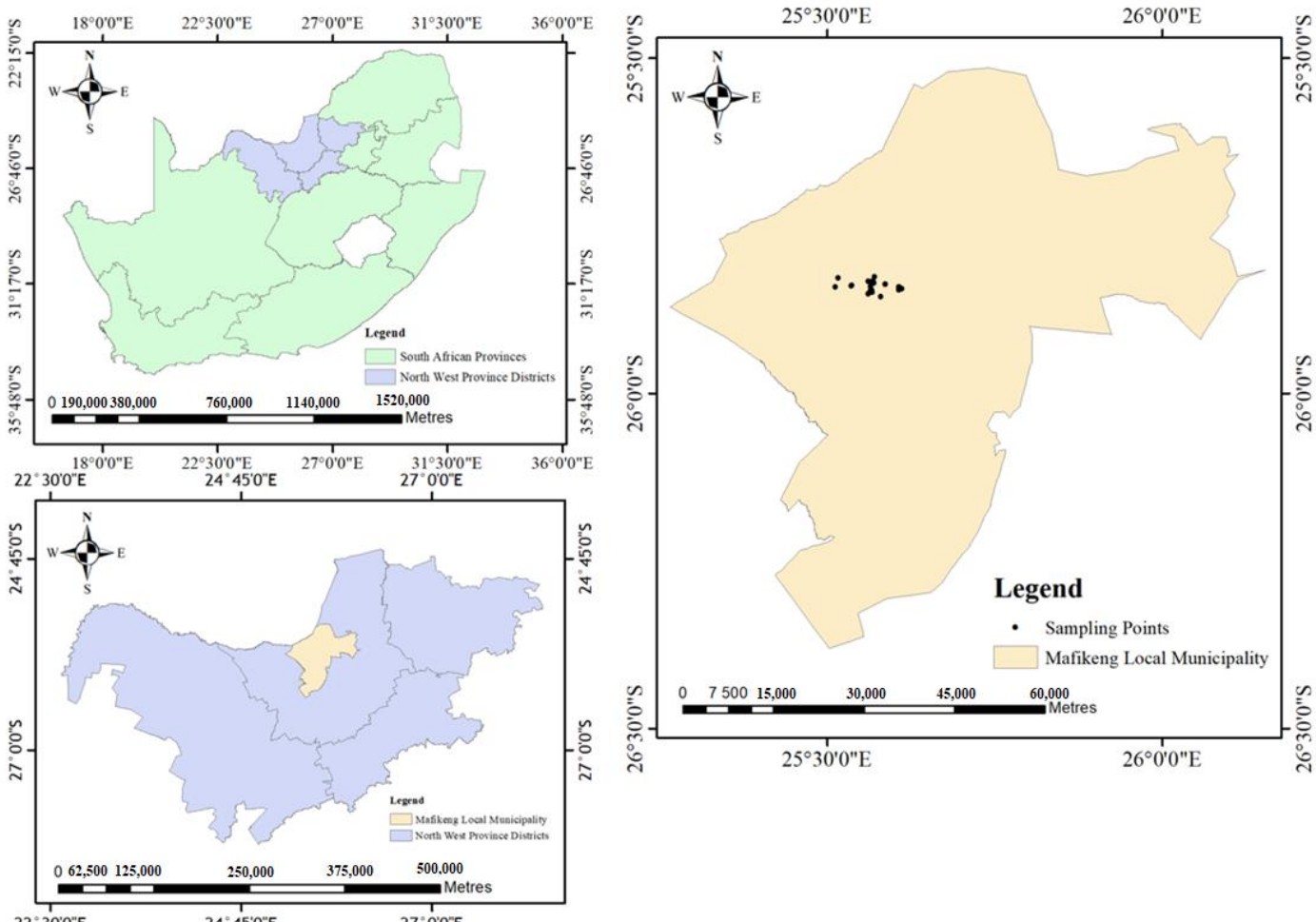

**Figure 1.** Map showing the GPS sampling points (not all sampling is shown as they overlap due to closeness).

In the study area, groundwater recharge is a diffused process that occurs through the direct movement of water from the land surface, resulting from precipitation that infiltrates and percolates through the unsaturated zones. The climate of the study area is influenced by two ecological regions—namely, the Highveld and the Southern Kalahari. The Highveld can be described as sub-humid, with low to moderate relief from the western side towards the eastern landscapes. The study area is semi-arid and records a mean annual rainfall of

400 mm. The area experiences mean summer temperatures, with the maximum in January at approximately 35 °C. Winters can be cool, with mean nighttime temperatures in June sometimes below 0 °C. The rate of evaporation exceeds that of precipitation. The relative humidity is between 64% and 66% in February and about 28–32% in July. Having a flat topography, the study area is underlain by a low to medium drainage density. Surface water is limited, and groundwater has become the only reliable water supply.

## 2.2. Sample Preparation

A total of 30 groundwater/borehole water samples were collected in July and October 2020 around the rural villages located in Mafikeng, namely, Dibate, Lokaleng, Moletsa-mongwe, Lekung, Airport View and Seweding. The water samples were collected by allowing the water from the borehole pump to flow freely for two minutes, then directly into the sample bottle, and then the bottle cap was replaced carefully. The water samples were collected in polyethylene bottles of 500 mL and carried to the alpha/beta spectrometry laboratory at the Centre for Applied Radiation Science and Technology (CARST) at North West University, Mafikeng campus, South Africa. Precautions were taken not to introduce contaminants into the water samples. The depth of the groundwater was between 40 m and 120 m, with a mean depth of 60 m.

The samples were grouped into 3 sets: the first set was group A, with 10 samples (S1–S10) from Dibate; the second set was group B, with 10 samples (S11–S20) from Lokaleng village; and the third was group C, with 10 samples (SM1–SM10) from Moletsamongwe, Lekung, Airport View and Seweding, located near Mmabatho unit 14 and Lokaleng village. The samples were collected at different locations (households) within the villages, separated by a distance of 5 km. The global positioning system (GPS) was used to record the coordinates of each point of the sample collection in terms of elevation above mean sea level, longitude and latitude.

Activity concentrations were measured using the direct method through liquid scintillation counting with the PerkinElmer® Wallac 1220 Quantulus (Ultra Low-Level Liquid Scintillation Spectrometer). The direct method allows for the groundwater samples to be mixed directly with the Ultima Gold uLLT cocktail. Prior to mixing with the cocktail, the groundwater samples were filtered using 90 mm filter papers (413 batch number). The 10 mL of the filtered water samples were placed in 20 mL polyethylene vials and then mixed with the organic solvent (Ultima Gold uLLT) cocktail from Perkin Elmer at a ratio of 10 mL to 10 mL (1:1). The 30 samples were prepared accordingly and placed between two photomultiplier tubes in the spectrometer for analysis.

## 2.3. Equipment

We used the Perkin Elmer Quantulus 1220 ultra-low-level liquid scintillation counter, (shown in Figure 2), to measure the activity concentration of two isotopes: tritium ($^3$H), a low-energy beta emitter, and radiocarbon ($^{14}$C), a high-energy beta emitter. Two protocols were created for both radiocarbon and tritium, respectively. The configuration selected for radiocarbon was $^{14}$C (high energy beta), and for tritium samples, it was $^3$H (low energy beta). The acquired spectra for the 2 protocols were opened and analysed using the Easy View (EV) software on the equipment computer. This instrument has its own background reduction system around the vial chamber, which consists of both an active and passive shield. The passive shield is made of lead, cadmium and copper, whereas the active shield is based on mineral oil around the vial chamber. The detection system converts incident radiation energy into fluorescence in an organic scintillator cocktail with a linear energy response and has advantages due to its high sensitivity and reproducibility. Organic scintillators are preferred for gross alpha/beta spectroscopy because of their hydrogen content [12].

The liquid scintillation counter was calibrated using both CPM/DPM counting mode and spectrum plot mode, as described by [13]. In CPM/DPM counting mode, an external radiation source was used to calibrate the extinction level, whereas the spectrum plot

mode was used to analyze the spectrum of the external radiation source. The internal standard method was used to establish the counter detector efficiency due to the low-level radioactivity [14]. Two commercial standards with known activities, a tritium standard (253,400.00 DPM $^3$H) and a radiocarbon standard (123,600.00 DPM $^{14}$C) from Perkin Elmer, were used to evaluate the detector efficiency as follows:

$$\text{Efficiency } \varepsilon = \frac{Count\ rate}{standard\ activity} \tag{1}$$

This implies;

$$\varepsilon = (R_{st} - R_b)/A_{st} \tag{2}$$

where $R_{st}$ is the tritium/radiocarbon standard count rate (cps), $R_b$ is the background aliquot count rate, in counts per second (cps), and $A_{st}$, is the activity of the tritium/radiocarbon standard. The calculated efficiency for tritium was 0.61 (61%) and 0.92 (92%) for radiocarbon. The Minimum Detectable Activity (*MDA*) was also evaluated using the Currie formula [12,15].

$$MDA\ (Bq/L)\ =\ \frac{L_d}{\varepsilon \times t_b \times V} \tag{3}$$

where,

$$detection\ limit\ L_d\ counts\ =\ 2.71 + 4.65\sqrt{R_b \times t_b} \tag{4}$$

$R_b$ is the background count rate (cps), $t_b$ is the counting time of the background sample (s), which was 720 min (240 min $\times$ 3), $\varepsilon$ is the efficiency calculated in Equation (2), and $V$ is the volume of the sample, which was 20 mL. Hence,

$$MDA\left(\frac{Bq}{L}\right) = \frac{2.71 + 4.65\sqrt{R_b \times t_b}}{\varepsilon \times t_b \times V}. \tag{5}$$

The *MDA* was calculated to be 0.58 Bq/L for tritium samples and 0.33 Bq/L for radiocarbon samples.

The samples were counted for 720 min (240 min $\times$ 3 cycles each). The spectra were evaluated by the computer software programme 1224-534 Easy View, which is a Windows 95, 98 NT 4.0 spectrum analysis software for Quantulus raw spectrum display and processing [16]. The activity concentration of $^3$H/$^{14}$C was calculated using the following formula:

$$A_s\ (Bq/L) = \frac{R_s - R_b}{\varepsilon \times V_s} \tag{6}$$

where $A_s$ is the activity concentration of $^3$H/$^{14}$C (Bq/L), $R_s$ is the sample count rate (cps), $R_b$ is the background count rate (cps), $\varepsilon$ is the calculated efficiency, and $V_s$ is the sample volume (20 mL). The obtained activity concentrations were used to estimate the annual effective dose (AED) using the following equation [17]:

$$AED\ (\mu Sv/y) = A_s\ (Bq/L) \times DC\ (\mu Sv/Bq) \times WCR\ (L/Y) \tag{7}$$

where *DC* is the dose coefficient and *WCR* is the water consumption rate per year. The lifetime cancer risk *(LTCR)* associated with the consumption of groundwater was also calculated using Equation (8) [18].

$$LTCR = A_s \times WCR \times LT \times CRC \tag{8}$$

where *WCR* is the water consumption rate taken as 731 L y$^{-1}$ for adults; *LT* is the average lifetime (year) for individuals, which is 58.6 years for males and 65.0 years for females according to the Statistics South Africa release (Mid-year estimates 2020) [19]; and *CRC* is the cancer risk coefficient given by the Environmental Protection Agency [20].

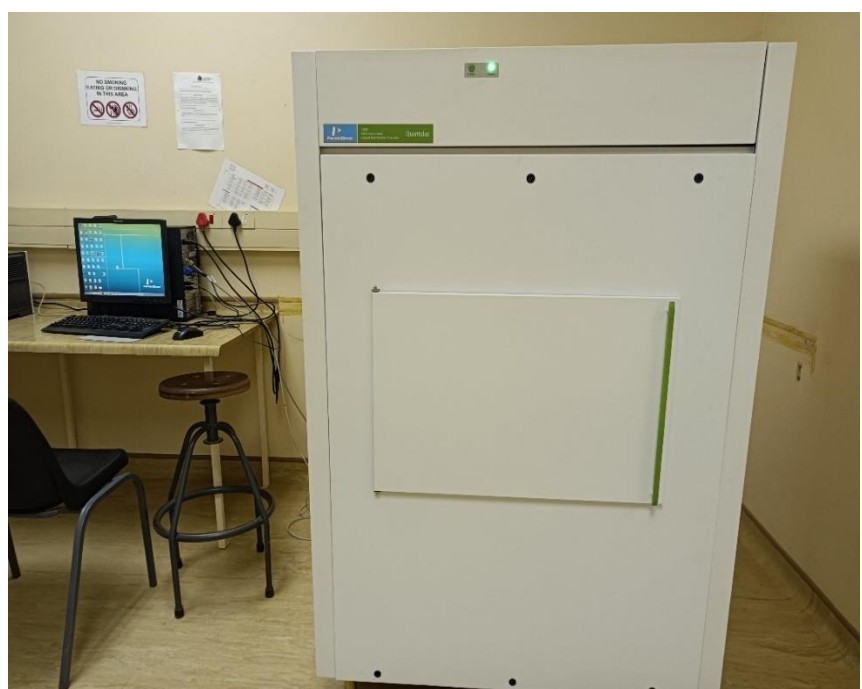

**Figure 2.** Perkin Elmer 1220 ultra-low-level liquid scintillation counter at the Center for Applied Radiation Science and Technology (CARST), North-West University (NWU), Mafikeng Campus, South Africa.

## 3. Results

Tables 1 and 2 give activity concentration values of tritium and radiocarbon, respectively, for the groundwater samples analysed, whereas Figure 3 depicts the annual effective dose rate for tritium and radiocarbon in the groundwater samples collected in the respective villages. Moreover, Figures 4–7 show the respective estimated lifetime cancer risk (LTCR) mortality and morbidity for tritium and radiocarbon in the collected groundwater samples from the various villages.

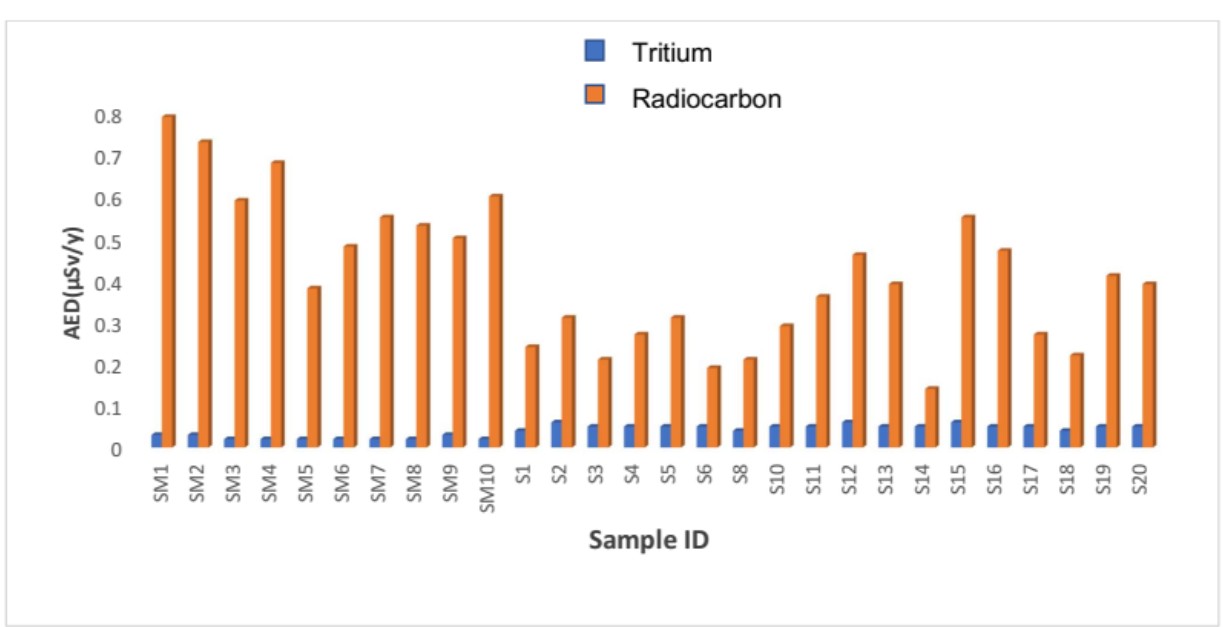

**Figure 3.** Annual effective dose rate for tritium and radiocarbon in the groundwater samples collected.

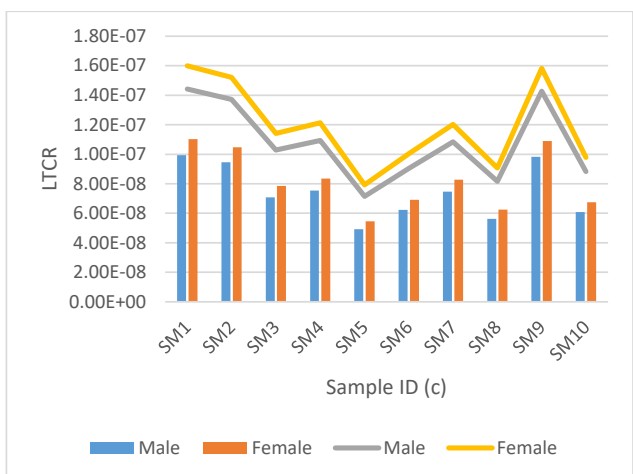

**Figure 4.** Estimated lifetime cancer risk (LTCR) mortality and morbidity for tritium are represented by the clustered bar graph and stacked line bar, respectively. This is for groundwater samples from Moletsamongwe, Lekung, Airport view and Seweding.

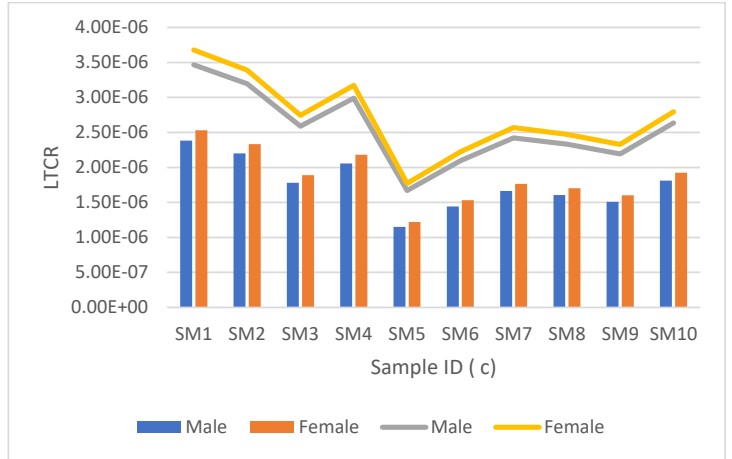

**Figure 5.** Lifetime cancer risk (LTCR) mortality vs. morbidity between men and women for radiocarbon is represented by the clustered bar graph and stacked line bar, respectively. This is for groundwater samples from Moletsamongwe, Lekung, Airport view and Seweding.

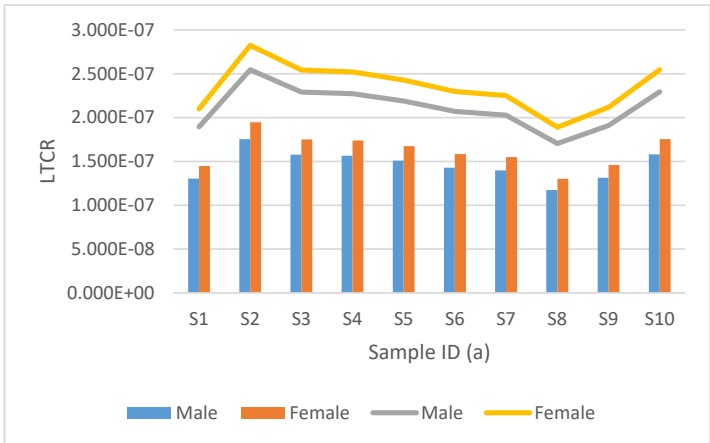

**Figure 6.** Estimated lifetime cancer risk (LTCR) mortality and morbidity for tritium are represented by the clustered bar graph and stacked line bar, respectively. This is from groundwater samples collected from Dibate village.

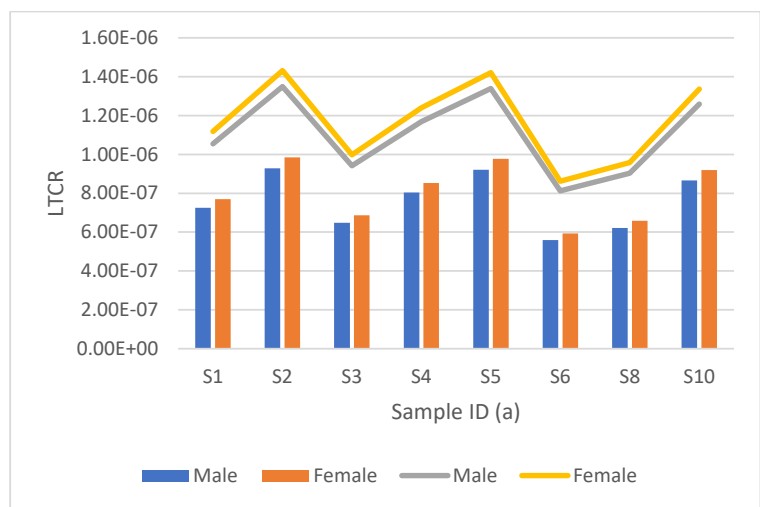

**Figure 7.** Lifetime cancer risk (LTCR) mortality vs. morbidity between men and women for radio-carbon is represented by the clustered bar graph and stacked line bar, respectively. This is from groundwater samples collected from Dibate village.

**Table 1.** Tritium ($^3$H) activity concentrations for the groundwater samples.

| Radioisotope | Sample ID | Crate(cps) | Cerror (cps) | Activity (Bq/L) | Activity Error |
|---|---|---|---|---|---|
| $^3$H | Standard | 1663.35 | 0.35 | | |
| $^3$H | Background | 0.10 | 0.00 | | |
| **Samples collected from Dibate village** | | | | | |
| $^3$H | S1 | 0.14 | 0.00 | 3.23 | 0.00 |
| $^3$H | S2 | 0.15 | 0.00 | 4.34 | 0.01 |
| $^3$H | S3 | 0.15 | 0.00 | 3.90 | 0.01 |
| $^3$H | S4 | 0.15 | 0.00 | 3.87 | 0.01 |
| $^3$H | S5 | 0.14 | 0.00 | 3.73 | 0.01 |
| $^3$H | S6 | 0.14 | 0.00 | 3.53 | 0.01 |
| $^3$H | S7 | 0.14 | 0.00 | 3.46 | 0.01 |
| $^3$H | S8 | 0.13 | 0.00 | 2.90 | 0.00 |
| $^3$H | S9 | 0.14 | 0.00 | 3.25 | 0.00 |
| $^3$H | S10 | 0.15 | 0.00 | 3.91 | 0.01 |
| | **Mean Activity Concentration** | | | **3.61** | **0.01** |
| **Samples collected from Lokaleng village** | | | | | |
| $^3$H | S11 | 0.15 | 0.00 | 4.05 | 0.01 |
| $^3$H | S12 | 0.15 | 0.00 | 4.35 | 0.01 |
| $^3$H | S13 | 0.14 | 0.00 | 3.71 | 0.01 |
| $^3$H | S14 | 0.15 | 0.00 | 3.93 | 0.01 |
| $^3$H | S15 | 0.15 | 0.00 | 4.28 | 0.01 |
| $^3$H | S16 | 0.14 | 0.00 | 3.77 | 0.01 |
| $^3$H | S17 | 0.14 | 0.00 | 3.74 | 0.01 |
| $^3$H | S18 | 0.14 | 0.00 | 3.38 | 0.01 |
| $^3$H | S19 | 0.15 | 0.00 | 3.93 | 0.01 |
| $^3$H | S20 | 0.14 | 0.00 | 3.46 | 0.01 |
| | **Mean Activity Concentration** | | | **3.86** | **0.01** |

**Table 1.** *Cont.*

| Radioisotope | Sample ID | Crate(cps) | Cerror (cps) | Activity (Bq/L) | Activity Error |
|---|---|---|---|---|---|
| **Samples collected from Moletsamongwe, Lekung, Airport View and Seweding** | | | | | |
| $^3$H | SM1 | 0.13 | 0.00 | 2.46 | 0.03 |
| $^3$H | SM2 | 0.13 | 0.00 | 2.34 | 0.03 |
| $^3$H | SM3 | 0.12 | 0.00 | 1.75 | 0.02 |
| $^3$H | SM4 | 0.12 | 0.00 | 1.86 | 0.02 |
| $^3$H | SM5 | 0.11 | 0.00 | 1.22 | 0.02 |
| $^3$H | SM6 | 0.12 | 0.00 | 1.54 | 0.02 |
| $^3$H | SM7 | 0.12 | 0.00 | 1.85 | 0.02 |
| $^3$H | SM8 | 0.12 | 0.00 | 1.39 | 0.02 |
| $^3$H | SM9 | 0.13 | 0.00 | 2.43 | 0.02 |
| $^3$H | SM10 | 0.12 | 0.00 | 1.50 | 0.02 |
| | **Mean Activity Concentration** | | | **1.83** | **0.02** |

**Table 2.** Radiocarbon ($^{14}$C) activity concentrations for the groundwater samples.

| Radioisotope | Sample ID | Crate(cps) | Cerror (cps) | Activity (Bq/L) | Activity Error |
|---|---|---|---|---|---|
| $^{14}$C | Standard | 1884.70 | 0.37 | | |
| $^{14}$C | Background | 0.07 | 0.00 | | |
| **Samples collected from Dibate village** | | | | | |
| $^{14}$C | S1 | 0.08 | 0.00 | 0.57 | 0.01 |
| $^{14}$C | S2 | 0.09 | 0.00 | 0.73 | 0.01 |
| $^{14}$C | S3 | 0.08 | 0.00 | 0.51 | 0.01 |
| $^{14}$C | S4 | 0.09 | 0.00 | 0.63 | 0.01 |
| $^{14}$C | S5 | 0.09 | 0.00 | 0.72 | 0.01 |
| $^{14}$C | S6 | 0.08 | 0.00 | 0.44 | 0.01 |
| $^{14}$C | S7 | 0.08 | 0.00 | >MDA | >MDA |
| $^{14}$C | S8 | 0.08 | 0.00 | 0.49 | 0.01 |
| $^{14}$C | S9 | 0.08 | 0.00 | >MDA | >MDA |
| $^{14}$C | S10 | 0.09 | 0.00 | 0.68 | 0.01 |
| | **Mean Activity Concentration** | | | **0.59** | **0.01** |
| **Samples collected from Lokaleng village** | | | | | |
| $^{14}$C | S11 | 0.09 | 0.00 | 0.84 | 0.01 |
| $^{14}$C | S12 | 0.09 | 0.00 | 1.08 | 0.02 |
| $^{14}$C | S13 | 0.09 | 0.00 | 0.92 | 0.01 |
| $^{14}$C | S14 | 0.08 | 0.00 | 0.34 | 0.01 |
| $^{14}$C | S15 | 0.10 | 0.00 | 1.30 | 0.02 |
| $^{14}$C | S16 | 0.09 | 0.00 | 1.11 | 0.02 |
| $^{14}$C | S17 | 0.09 | 0.00 | 0.64 | 0.01 |
| $^{14}$C | S18 | 0.08 | 0.00 | 0.52 | 0.01 |
| $^{14}$C | S19 | 0.09 | 0.00 | 0.97 | 0.01 |
| $^{14}$C | S20 | 0.09 | 0.00 | 0.91 | 0.01 |
| | **Mean Activity Concentration** | | | **0.83** | **0.01** |

**Table 2.** *Cont.*

| Radioisotope | Sample ID | Crate(cps) | Cerror (cps) | Activity (Bq/L) | Activity Error |
|---|---|---|---|---|---|
| Samples collected from Moletsamongwe, Lekung, Airport View and Seweding | | | | | |
| $^{14}C$ | SM1 | 0.11 | 0.00 | 1.86 | 0.03 |
| $^{14}C$ | SM2 | 0.11 | 0.00 | 1.72 | 0.02 |
| $^{14}C$ | SM3 | 0.10 | 0.00 | 1.39 | 0.02 |
| $^{14}C$ | SM4 | 0.10 | 0.00 | 1.61 | 0.02 |
| $^{14}C$ | SM5 | 0.09 | 0.00 | 0.90 | 0.01 |
| $^{14}C$ | SM6 | 0.10 | 0.00 | 1.13 | 0.02 |
| $^{14}C$ | SM7 | 0.10 | 0.00 | 1.30 | 0.02 |
| $^{14}C$ | SM8 | 0.10 | 0.00 | 1.25 | 0.02 |
| $^{14}C$ | SM9 | 0.10 | 0.00 | 1.18 | 0.02 |
| $^{14}C$ | SM10 | 0.10 | 0.00 | 1.42 | 0.02 |
| | **Mean Activity Concentration** | | | **1.38** | **0.02** |

## 4. Discussion

### 4.1. Activity Concentrations

Table 1 shows the values of activity concentration of tritium in the groundwater samples S1–S10, S11–S20 and SM1–SM10 collected from Dibate, Lokaleng and Moletsamongwe, Lekung, Airport View and Seweding villages, respectively, whereas Table 2 reveals the values of activity concentration of radiocarbon for the various respective villages. The activity concentrations ranged from $2.90 \pm 0.002$ Bq/L to $4.34 \pm 0.01$ Bq/L, with an average of $3.61 \pm 0.01$ Bq/L for samples collected from Dibate village, and for Lokaleng village, they ranged from $3.46 \pm 0.01$ Bq/L to $4.34 \pm 0.01$ Bq/L with an average of $3.86 \pm 0.01$ Bq/L. The samples SM1–SM10 have a lower activity concentration for tritium, with an average of $1.83 \pm 0.02$ Bq/L. These samples were from Moletsamongwe, Lekung, Airport View and Seweding villages. Furthermore, from Table 2, the activity concentrations of radiocarbon ranged from $0.44 \pm 0.01$ Bq/L to $0.73 \pm 0.01$ Bq/L, with an average of $0.59 \pm 0.01$ Bq/L for samples from Dibate and for Lokaleng village samples, it had an average of $0.86 \pm 0.01$ Bq/L, with the activity ranging from $0.34 \pm 0.01$ Bq/L to $1.29 \pm 0.02$ Bq/L. The highest radiocarbon activity concentration in the obtained results is $1.86 \pm 0.03$ Bq/L from sample SM1, one of the samples collected from Moletsamongwe, Lekung, Airport View and Seweding villages. The samples have an average activity concentration of $1.38 \pm 0.02$ Bq/L. These disparities might be due to the geological variations in the studied areas.

### 4.2. Annual Effective Dose Rate (AED)

Figure 3 shows the estimated AED of $^3H$ and $^{14}C$ in each of the groundwater samples that were collected. From Lokaleng village, the results ranged from $0.04453 \pm 0.00739$ μSv/y to $0.05723 \pm 0.00739$ μSv/y, with an average AED of $0.05079 \pm 0.00739$ μSv/y. The sample S12 was found to have the highest AED, but S15 had the lowest. The AED for groundwater samples from Moletsamongwe, Lekung, Airport View and Seweding ranged from $0.01602 \pm 0.02265$ μSv/y to $0.03198 \pm 0.02265$ μSv/y, with an average of $0.02413 \pm 0.02265$ μSv/y. The sample SM1 was found to have the highest AED, but SM5 had the lowest. The tritium annual effective dose rate (AED) difference in Figure 3 results from the difference in activity concentrations because the dose rate is directly affected by the activity concentration; when the activity was high, so was the AED, and when the activity was low, the AED was also low. However, the activity concentrations of the borehole waters collected differ depending on the physicochemical and geochemical conditions and the geological formation of the soil and bedrock in each area [21]. The activity concentrations and annual effective dose rates from the studied villages vary for the samples analysed

for both tritium and radiocarbon, and that might be due to the geological variations in the studied areas.

Moreover, the annual effective dose rate (AED) for $^{14}$C groundwater samples from Dibate village ranged from $0.18514 \pm 0.00612$ μSv/y to $0.30742 \pm 0.00612$ μSv/y with an average of $0.251404 \pm 0.00612$ μSv/y. The sample S2 was found to have the highest AED, but S6 had the lowest. The estimated AED for $^{14}$C groundwater samples collected from Lokaleng village ranged from $0.14318 \pm 0.00739$ μSv/y to $0.55051 \pm 0.00739$ μSv/y, with an average AED of $0.36604 \pm 0.00739$ μSv/y. The sample S15 was found to have the highest AED, but S14 had the lowest. The AED for groundwater samples from Moletsamongwe, Lekung, Airport View and Seweding ranged from $0.47761 \pm 0.02265$ μSv/y to $0.78994 \pm 0.02265$ μSv/y, with an average of $0.58309 \pm 0.02265$ μSv/y. The sample SM1 was found to have the highest activity concentration, but SM5 had the lowest. The estimated dose values from each of the studied villages were found to be below the recommended annual dose value of 100 μSv as per guidelines for drinking water quality set by the World Health Organization [22]. Hence, the local Mafikeng groundwater is radiologically safe for human consumption.

The mortality and morbidity rates for the lifetime cancer risk (LTCR) as a result of consuming the studied groundwater (borehole water) were calculated for adults: males and females, with an average lifetime expectancy of 58.6 years for males and 65.0 years for females, respectively. The comparisons for LTCR mortality and morbidity between men and women are depicted graphically in Figures 4–7.

Figures 4 and 5, show the LTCR mortality and morbidity for groundwater samples analysed for tritium for data collected at Moletsamongwe, Lekung, Airport View and Seweding villages.

Figures 6 and 7, show the LTCR results for radiocarbon, for data collected from Dibate village. The mortality rate for males varied from $1.1743 \times 10^{-7}$ to $1.7543 \times 10^{-7}$, with a mean average of $1.4610 \times 10^{-7}$, and the mortality rate for females varied from $1.3026 \times 10^{-7}$ to $1.9460 \times 10^{-7}$ with an average of $1.6206 \times 10^{-7}$. The morbidity rate for males ranged from $1.7095 \times 10^{-7}$ to $2.5461 \times 10^{-7}$, with a mean average of $2.1204 \times 10^{-7}$, while the morbidity rate for females ranged from $1.8904 \times 10^{-7}$ to $2.8241 \times 10^{-7}$ with a mean average of $2.3519 \times 10^{-7}$. It can be observed that S8 had the lowest LTCR in both mortality and morbidity for males and females, but S2 had the highest LTCR. The slight difference in the LTCR results from different activity concentrations and different sampling locations; some locations have a high tritium activity concentration, but some have a low concentration.

Samples from Moletsamongwe, Lekung, Airport View and Seweding are shown in Figures 4 and 5. The mortality rate for males ranged from $4.4925 \times 10^{-8}$ to $9.9136 \times 10^{-8}$, with a mean average of $1.7416 \times 10^{-8}$, and the mortality rate for females varied from $5.4624 \times 10^{-8}$ to $1.1021 \times 10^{-7}$, with an average of $8.2259 \times 10^{-8}$. The morbidity rate for men ranged from $7.1460 \times 10^{-8}$ to $1.4419 \times 10^{-7}$, with a mean average of $1.0763 \times 10^{-7}$, while the morbidity rate for women ranged from $7.9270 \times 10^{-8}$ to $1.5994 \times 10^{-7}$, with a mean average of $1.193 \times 10^{-7}$. SM15 had the lowest LTCR in both mortality and morbidity for males and females, but SM1 had the highest LTCR. Figure 7 shows the results from the data collected from Moletsamongwe, Lekung, Aiport View and Seweding. The average male mortality rate was found to be $1.76067 \times 10^{-6}$ and $2.5588 \times 10^{-6}$ for morbidity, while the average female mortality rate was found to be $1.8682 \times 10^{-6}$ and $2.7150 \times 10^{-6}$ for morbidity. SM5 had the lowest LTCR in both mortality and morbidity for males and females, but SM1 had the highest LTCR.

It can be observed from Figure 7 that women have a high lifetime cancer risk of mortality and morbidity in all the samples analysed for both tritium and radiocarbon. Women have a longer life expectancy than men, which explains why women have a slightly higher lifetime cancer risk than men. When comparing the sets of results for groundwater samples analysed for radiocarbon and tritium, it was evident that the samples analysed for radiocarbon had a lifetime cancer risk (morbidity and mortality) higher than the ones analysed for tritium. Although the samples analysed for radiocarbon had a lower

activity concentration than the samples with tritium, radiocarbon has a greater cancer risk coefficient compared to tritium, which results in an overall high lifetime cancer risk. However, the lifetime cancer risk values for this study were lower than the radiological cancer risk limit of $1.6300 \times 10^{-3}$ set by the United Nations Scientific Committee on the Effects of Atomic Radiation (UNSCEAR 2008 report) [23]. Therefore, it can be concluded that the borehole water from these villages is safe to drink.

## 5. Conclusions

The estimated dose values from each of the studied villages were found to be below the recommended annual dose value of 100 μSv as per guidelines for drinking water quality set by the World Health Organization. The radiological hazards data in this study show that the potential radioactive risks caused by tritium and radiocarbon were within acceptable limits in the study area. There was no health risk posed to the public by drinking the water from these sources, which could help put the public at ease regarding any concerns they might have about the effects of radioactivity in groundwater. Although the radiological doses in this study are well within the limits, according to ALARA, no dose should be acceptable if it can be avoided. Therefore, radiological studies are important from time to time to monitor the quality of the villages' aquifers, which are being abstracted for consumption by mankind. There are no similar studies that were previously performed for the analysis of tritium and radiocarbon in borehole waters in these specific villages.

This study showed that the village boreholes are radiologically safe for human consumption. Since this study is the first of its kind in the study area, our results would serve as the baseline for natural background levels of $^3$H and $^{14}$C in the area. Furthermore, the study would serve as a valuable reference for the scientific community and policymakers in the future.

**Author Contributions:** Conceptualization, O.E.D.; methodology, O.E.D.; validation, O.E.D., J.M. and M.M.; formal analysis, O.E.D.; investigation, O.E.D. and J.M.; data curation, O.E.D. and J.M.; writing—original draft preparation, O.E.D.; writing—review and editing, O.E.D., J.M. and M.M.; visualization, O.E.D.; supervision, M.M. All authors have read and agreed to the published version of the manuscript.

**Funding:** This research received no external funding.

**Data Availability Statement:** No new data were created or analyzed in this study. Data sharing is not applicable to this article.

**Acknowledgments:** The authors would like to thank the different villagers who gave us access to their homesteads and provided us with water that was used for experiments.

**Conflicts of Interest:** The authors declare no conflict of interest.

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
