# Peer review of "Environmental Monitoring of Tritium (3H) and Radiocarbon (14C) Levels in Mafikeng Groundwater Using Alpha/Beta Spectrometry"

_water, doi:10.3390/w15112037_

Round 1

Reviewer 1 Report (New Reviewer)

Review

Abstract

Maybe add “South Africa”  just after Mafikeng (Line 1) for people not aware of Mafikeng location or just reading the abstract without continuing with the whole paper

Have to be rewritten. Too many words duplicated (ex. Line 9 : availability of available water resources…).

It seems to the reviewer that the abstract cannot be considered as an abstract, since it is just the presentation of the results without any summary of objectives, discussion and so on, except the very last sentence where the key words appear. Moreover, why emphasizing the alpha/beta spectrometer in the title and abstract? Is there a specific reason? No precision on this point is given.

Introduction

Line 31: no need for “also”

Lines 34-35: I do not understand why there is “such as borehole” added to the sentence. I think stop after groundwater resources will be enough. 

Lines 40-42: a little bit repetitive….

Line 45: Since the paper seems to focus on radionuclides in GW, maybe this sentence should perhaps have been developed a little further. Indeed, the possible pollution by radionuclides is of utmost importance when dealing with natural radioactive tracers, the former having to be distinguished from the latter.

Lines 52-70: some more pertinent references would have been appreciated, mainly for the use of 3H and 14C in groundwater studies. There is a huge bibliography published by the most internationally well-known scientists in this field.

Lines : I don't understand why there is a danger to human health with tritium and carbon-14 in the study area. There may be a danger if there is an input of anthropogenic 3H, such as from nuclear power plant discharges, which are the two most emitted isotopes into the environment from nuclear facilities in normal operation. There is no explanation here of the cause of any increase in 3H and/or C14... which is absolutely missing.

Lines 109-111: too general. Useless. 

It would be useful for the reader if the authors resumed the order of the points in the paragraph "Geological and hydrological frameworks". I can advise: geology ---> climatic and ecological zones, and associated precipitation ---> topography and weathering ---> trace elements.

Sample location and preparation

Nothing to add. Good description.

Line 155: The "Equipment" section is a bit long and seems to come from the presentation leaflet of the equipment used. However, it may be acceptable to detail this measuring instrument as it is mentioned in the title of the article, and therefore, its use should be important for the following.

This does not concern the cancer risk equations, which are very briefly described at the end of the introduction.

Lines 226-232: The presentation of the results is also very succinct, with relevant diagrams but little description. No details on the figure captions.

Perhaps the discussion is carried over into the discussion?

Discussion

Activity concentrations (Lines 272-290): this paragraph would have been better in the “Results” section because except the calculation of averages, it is only the description of the results and not a discussion.

Note in Line 279: there is a dot mission in the average 3H concentration (386 instead of 3.86).

Lines 325-329: this text would have been better placed in the figure captions of the graphs for an easier understanding of the “clustered bar graph” and the “stacked line graph”.

Lines 354-366 appear to be the real discussion…

Conclusions

The conclusion of this project is that groundwater does not constitute a health hazard for the populations in the study areas. In the absence of any particular discussion of radioactive discharges in the region or of a very specific geology, this conclusion seemed very quickly obvious to the reviewer when reading the article.

However, if the subject had been to build a monitoring network, which the authors claim would have been the first in the study area, the article could have gained much more from the measurements made.

No remarks

Author Response

See attached Table of corrections for Reviwer 1

Reviewer 2 Report (Previous Reviewer 2)

Please find the attached document herein.

Too many figures for same results, need shorten for the readers.

Author Response

See attached Table of Corrections for Reviewer 2

Round 2

Reviewer 1 Report (New Reviewer)

I thank the authors for the corrections made to the text.

However, in the introduction (lines 57-62), explanations of possible radionuclide pollution are given but remain insufficient: inputs from medical waste disposal or human activities, the latter not being detailed. And this is the main point of this document. 

I consider that the possible causes of groundwater pollution are not developed enough to understand the fundamental objectives of this paper. 

However, as I mentioned earlier, the authors want to present their measurements (of a very good scientific level) to set the stage for future environmental monitoring and database construction. In this case, the introduction should be revised, and the title should read "environmental monitoring" for a stronger impact. 

Author Response

Please see Corrections attached.

Reviewer 2 Report (Previous Reviewer 2)

The manuscript is now much improved. The authors have done a great job of responding to comments all over the article and improving the English as well. This manuscript will make a good contribution to the literature as a research article. Thank you for all your efforts. I recommend to accept this article to publish.

Moderate editing of English language is required.

Author Response

Please see attached Corrections

This manuscript is a resubmission of an earlier submission. The following is a list of the peer review reports and author responses from that submission.

Round 1

Reviewer 1 Report

Revision of the manuscript “Determination of tritium (3H) and radiocarbon (14C) levels in Mafikeng groundwater using alpha/beta spectrometry” by Dzimba et al.

I am sorry but I suggest to reject this manuscript. In my opinion, the authors lacks the understanding of the topic, in particular in hydrogeology, groundwater pollution and usage of isotopes in hydrogeology. Please find my general and specific comments below.

General comments:

1.      The abstract is too long. According to the “Guide for authors” an abstract shout be a single paragraph of about 200 words maximum. The abstract is twice that long.

2.      Introduction is the best part of this manuscript. It is well written.

3.      The authors stated that one of the objectives of this study is to estimate the annual effective dose rates. Unfortunately, the introduction does not include any information to this topic. The methodology of this estimation is not presented.

4.      The results comprise only figures without any description. The figures are described barely in the discussion.

5.      Presented results are not related to the introduction.

6.      The references list comprises modest number of ca. 12 publications.

Specific comments:

L8-9: no predicate in the first sentence. Please correct.

L115-144: add references

L315-320: give only initials, not your full names; quotation marks unnecessary

Reviewer 3 Report

This manuscript analyzes the radioactivity concentration of groundwater samples. Whether there were naturally occurring radioactive material in the study area in the past. Please add an explanation in the introduction.